# Secure Air Traffic Control at the Hub of Multiplexing on the Centrifugo-Pneumatic Lab-on-a-Disc Platform

**DOI:** 10.3390/mi12060700

**Published:** 2021-06-15

**Authors:** Jens Ducrée

**Affiliations:** School of Physical Sciences, Dublin City University, Glasnevin, Dublin 9, Ireland; jens.ducree@dcu.ie; Tel.: +353-1-700-5377

**Keywords:** centrifugal microfluidics, Lab-on-a-Disc, centrifugo-pneumatic flow control, integration, sample-to-answer automation, multiplexing, parallelization, reliability, tolerances, design-for-manufacture, digital twin, event triggering, timing

## Abstract

Fluidic larger-scale integration (LSI) resides at the heart of comprehensive sample-to-answer automation and parallelization of assay panels for frequent and ubiquitous bioanalytical testing in decentralized point-of-use/point-of-care settings. This paper develops a novel “digital twin” strategy with an emphasis on rotational, centrifugo-pneumatic flow control. The underlying model systematically connects retention rates of rotationally actuated valves as a key element of LSI to experimental input parameters; for the first time, the concept of band widths in frequency space as the decisive quantity characterizing operational robustness is introduced, a set of quantitative performance metrics guiding algorithmic optimization of disc layouts is defined, and the engineering principles of advanced, logical flow control and timing are elucidated. Overall, the digital twin enables efficient design for automating multiplexed bioassay protocols on such “Lab-on-a-Disc” (LoaD) systems featuring high packing density, reliability, configurability, modularity, and manufacturability to eventually minimize cost, time, and risk of development and production.

## 1. Introduction

There is a strong market need for providing rather complex bioanalytical testing capabilities that are currently carried out by well-trained personnel in professional facilities, for manifold decentralized use cases. Main applications are found in biomedical point-of-care and global diagnostics, liquid handling automation for the life sciences, process analytical techniques, and cell line development for biopharma, as well as monitoring the environment, infrastructure, industrial processes, and agrifood. Since the early 1990s, microfluidic “Lab-on-a-Chip”, also—now quite synonymously—referred to as Micro Total Analysis Systems (“µTAS”), have been proposed to cater for these strong demands [1,2,3,4,5].

Since these pioneering days, centrifugal microfluidic technologies have played a significant role in the scientific community [6,7,8,9,10,11,12,13,14,15,16,17,18,19,20,21,22,23,24,25,26,27,28,29], and, even more, in commercially directed initiatives [30,31,32,33,34,35,36,37,38,39,40,41,42]; as common in pioneering technologies, many of these companies have experienced significant success, while others were absorbed or discontinued in the meantime. The development of such “Lab-on-a-Disc” (LoaD) systems involves the well-coordinated development of fluidic, manufacturing, instrumentation, software, detection, and, last but not least, bioassay capabilities. The latter two elements of the LoaD technology stack are often closely derived from established reagent kits, and optical or electrochemical sensing techniques. However, the degrees of freedom for fluidic layouts are significantly restricted, chiefly owing to the typical, single-use character of the carrier chip as the multi-branched channel networks can typically not be regenerated.

To still mitigate the cost, the disposable chips are usually produced in polymer materials by (compression-)injection molding, which is a well-established technique in optical data storage manufacturing, and thus often adopt geometries such as Compact Disc^TM^ (CD), Digital Video Disc (DVD), or Blu-ray formats (or sections thereof) with their standard 12 cm diameter, thickness of about 1.2 mm, and a central, 15 mm diameter hole for coupling to the spindle. The disc shape is also commensurate with the rotational symmetry of the centrifugal field. While centrifugal microfluidic devices have been implemented on many other shapes, e.g., circular segments, semicircle, standard (rectangular) microscope slides, and even tubes, for the sake of this work, the popular term “disc” will represent all types of chip formats attached to a rotor.

Similar to a CD player or Discman, LoaD systems usually feature a modular setup with a rather rugged, possibly portable, and widely autonomous instrument taking up the fluidic carrier chip, which is equipped with the spindle motor, control, detection, display, and communication units. For immersion with most point-of-use scenarios outside professional laboratory infrastructure, all (liquid or dry) reagents need to be (pre-)stored on the chip so that only the (bio-)sample needs to be loaded by the end-user.

While the physical representation, i.e., pits and lands, on optical storage media are of micron- and sub-micron size, the channels and chamber in microfluidic systems commonly need to handle volumes in the upper microliter range, at least on the inlet side, corresponding to milli- to centimeter lateral dimensions. Such multi-scale manufacture of the low-aspect-ratio discs imposes significant challenges on manufacturing and assembly, which need to be addressed separately for each fluidic design and fabrication technology. Furthermore, as with other conventional Lab-on-a-Chip technologies, the development of LoaD applications goes through several stages, often starting from prototyping for initial proof-of-concept, pilot series, e.g., to establish sufficient statistics for fluidic and bioanalytical validation, and substantial optimization towards commercially viable mass production.

Each fabrication scheme, along with the scale-up, e.g., precision milling in prototyping and subsequent tool-based replication, involves restrictions on geometries, manufacturing tolerances, artefacts, and often entails considerable efforts for setting up the process [43]. Typically, any change in the fluidic layout, materials, manufacturing processes, detection concepts, instrumentation, software, reagents, and assay protocol may induce mutual interference. Consequently, diligently following Design-for-Manufacture (DfM) guidelines and proper interface management are thus key for reaching high Technology Readiness Levels (TRLs) within reasonable techno-economical boundary conditions.

Most centrifugal microfluidic platforms operate in a batch-wise “stop-and-go” fashion, meaning that Laboratory Unit Operations (LUOs) for sample preconditioning, such as metering/aliquoting [44,45,46], removal of bioparticles [29,47,48,49,50,51], incubation, purification/concentration/extraction [52,53], homogenization [54,55], resuspension of dry, and mixing with liquid reagents [56,57,58,59] are gated by normally closed valves. Upon completion of each LUO, the liquid is forwarded to the next step. These centrifugal LUOs and their connected downstream detection methods have been surveyed before [7,8,12,19,20,21,22,23,24,25,26,27,60,61,62,63,64,65,66,67].

Techniques for such valving can be broadly distinguished by their retention and release mechanisms [68,69]. Initial LoaD concepts were mainly based on siphoning [31,32] and capillary burst valves [6], typically requiring the precise definition of geometrical features or surface coatings. Concepts based on sacrificial materials, for instance, wax [70,71,72,73], membranes [74], or stick-packaging [75], provide a liquid and vapor barrier, thus making them suitable for longer-term onboard storage of liquid buffers or reagents; yet, they either require an externally powered unit [67,76] or very high, often poorly defined release rates. Rudimentary centrifugo-pneumatic (CP) flow control mechanisms were introduced [77,78], which tend to require large, rotationally induced pressure heads for opening; therefore, they are primarily suitable for valving the penultimate LUO in a liquid handling sequence.

Subsequent work refined the release mechanism of these CP valving techniques by venting their compression chamber. Other than removing the seal by instrument-based units such as lasers [74] or xurography [79], (water) dissolvable films (DFs) covering outlets that are pneumatically coupled to the compression chamber were opened by ancillary liquids [80]. With this strategy, complex, highly multiplexed networks offering “handshake” as well as logical flow control, such as AND, OR, and IF conditionals, could be developed, which operate widely independently of the spin protocol [49,81,82,83,84].

This paper investigates key design features gearing centrifugo-pneumatically coupled valving networks towards larger-scale integration (LSI) of parallelized, multi-step bioassay panels. Note that the notion of fluidic LSI has previously been used by Quake et al. [85,86] in a different context of a pneumatically multiplexed liquid handling on a stationary, elastomeric platform, which is to be distinguished from the centrifugal technology be presented here. For the first time, the approach presented here also factors in experimental tolerances to deliver high predictability for batch-mode operated microfluidic LoaD platforms [68]. In pursuit of this novel objective, a “digital twin” [87,88], i.e., a model-based representation that serves as the virtual counterpart of a physical object or process [68,87,88], is developed for optimizing the fluidic performance, robustness, packing density, and manufacturability of rotationally controlled valving schemes for LoaD platforms.

First, the basic principles of centrifugal microfluidic liquid handling are briefly reviewed to establish the mathematical framework of the digital twin. Then a set of performance metrics is introduced along which fluidic layouts can be systematically optimized for given design objectives. The next section elaborates on the fluidic multiplexing in the context of real space and band widths in the frequency domain induced by standard deviations experimental input parameters. The method is illustrated by an exemplary bioassay. In the next section, the digital twin concept is applied to advanced flow control schemes allowing enhanced multiplexing and flexible timing before concluding.

## 2. Basics of Rotational Flow Control

### 2.1. Centrifugal Field

According to classical mechanics, an object of (uniform) density $\varrho_{\mathrm{part}}$ and volume V with a center of mass located at a distance *R* from the axis of rotation experiences a centrifugal field density
(1)$$f_{\omega }=F_{\omega }/V=\varrho_{\mathrm{part}}\cdot R\cdot \omega^{2}$$
at the angular spin rate ω=2π⋅ν. Note that in typical microfluidic scenarios, a (bio-) particle suspended in a liquid of density ϱ experiences buoyancy, so the $\varrho_{\mathrm{part}}$ in $f_{\omega }$ (1) would refer to the difference between the densities $\varrho_{\Delta }=\varrho_{\mathrm{part}}-\varrho$ of the object and its surrounding medium.

While not relevant to the immediate context of this paper focusing on valving, the rotationally induced Coriolis (pseudo) force $|f_{v}|=2\varrho \cdot \omega \cdot v$ acts on objects traveling at a speed v=ω⋅r in the non-inertial frame spinning at ω. In common LoaD systems, ${\vec{f}}_{v}$ is directed opposite to the direction of motion in the plane of the disc, and has been used for agitating advective mixing [57,59] and routing [89].

### 2.2. Pressure Contributions

Rather than forces, the response of liquids as continuous media to a field $f_{\omega }$ (1) is better captured by pressures p=F/A, i.e., force F per unit (cross-sectional) area A. In the context of the essentially hydrostatic approximation underlying the mainstay of this paper, flow-induced effects relating to the viscosity of the liquid are neglected. A valve is represented by a geometrical structure Γ which is composed of chambers and interconnecting channels holding a liquid of total volume $U_{0}$.

On LoaD systems, the distribution of the (contiguous) liquid Λ(𝜔) within Γ forms under the impact of the centrifugal pressure head
(2)$$p_{\omega }=\varrho \cdot \bar{r}\Delta r\cdot \omega^{2}$$
scaling with the square of the spin rate ω and the radial product
(3)$$\bar{r}\Delta r=\frac{1}{2}(r^{2}-r_{0}^{2})$$
composed of its mean radial position
(4)$$\bar{r}\Delta r=\frac{1}{2}(r^{2}-r_{0}^{2})$$
and its radial extension
(5)$$\Delta r=r-r_{0}$$
which is (radially) confined by its inner and outer menisci $r_{0}$ and r, respectively. For typical values $\varrho =10^{3}\;{\mathrm{kg}\;m}^{-3}$, $\bar{r}=3\;\mathrm{cm}$, and Δr=1 cm, spin rates ν=ω/2π=10 Hz and 50 Hz, rotationally induced pressure heads $p_{\omega }$ (2) of roughly 12 hPa and 300 hPa are obtained, the latter corresponding to about 1/3 of the standard atmospheric pressure $p_{\mathrm{std}}=1013.25\;\mathrm{hPa}$.

The net counterpressure in the axial z-direction opposing $p_{\omega }$ can be compounded to $p_{z}=p_{\leftarrow }-p_{\to }$ with (the gradients of) the pressures pointing parallel $p_{\to }$ and opposite $p_{\leftarrow }$ to the main (axial) direction of the channel. Based on the hydrostatic equilibrium
(6)$$p_{\omega }+p_{\to }=p_{\leftarrow }\Leftrightarrow p_{\omega }=p_{z}$$
and the continuity of liquid volume
(7)$$\frac{d}{dt}\int_{\Lambda }^{}dV=0$$
the two (radial) positions r and $r_{0}$ of its confining menisci, and thus the distribution Λ throughout Γ, can hence, either algebraically or numerically, be calculated.

The contributions $p_{\to }$ and $p_{\leftarrow }$ might be generated externally [11,74,76,83,90,91,92,93], e.g., by compressed air bottles, onboard pumps, or mechanical valves (corresponding to $p_{\leftarrow }\mapsto \infty$); for more common, rotationally controlled valving mainly investigated in the mainstay of this work, pneumatic pressures
(8)$$p_{V}=p_{0}\cdot \frac{V_{0}}{V}$$
linked to the compression of a gas volume enclosed by the liquid from their original size $V_{0}$ at the ambient atmospheric pressure $p_{0}$ to $V<V_{0}$, e.g., by mechanical work or temperature change. Through shrinking V, $p_{V}$ (8) may, theoretically, be made randomly large.

Capillary action is associated with a pressure
(9)$$p_{\Theta }=\frac{4\sigma }{D}\cdot \cos\Theta$$
governed by the diameter D of the channel (with a round cross section) and the contact angle Θ of its inner wall with a liquid of surface tension σ. have been implemented. Typical values D=100 µm, $\sigma \approx 72.8\times 10^{-3}\;{N\;m}^{-1}$ and Θ=120° yield $p_{\Theta }\approx 15\;\mathrm{hPa}$ (9), which corresponds to rather low spin rates in the regime of ν=ω/2π≈10 Hz.

### 2.3. Angular Acceleration

Certain LUOs, e.g., for mixing by chaotic advection [56] and or mechanical cell lysis [94] with obstacles or (magnetic) beads, benefit from rotational accelerations dv/dt=R⋅dω/dt. Assuming a solid disc of mass $m_{\mathrm{disc}}$ and radius $R_{\mathrm{disc}}$, and thus a moment of inertia $I_{\mathrm{disc}}\approx 0.5\cdot m_{\mathrm{disc}}\cdot R_{\mathrm{disc}}^{2}$ and resulting angular momentum $L_{\mathrm{disc}}=I_{\mathrm{disc}}\cdot \omega$, the steepness of the ramps $d\omega /dt=\tau_{\mathrm{spindle}}/I_{\mathrm{disc}}$ is limited by the maximum torque $\tau_{\mathrm{spindle}}$ of the spindle motor. Such angular accelerations might be approximated by an inertially induced pressure term
(10)$$p_{m}=\frac{\varrho \cdot U\cdot R\cdot d\omega /dt}{A}\leq \frac{\varrho \cdot U\cdot R\cdot \tau_{\mathrm{spindle}}}{A\cdot I_{\mathrm{disc}}}$$
for a fluid volume U of density ϱ traveling at a speed v=R⋅ω at a radial position R through a channel of cross-section A.

Note that LoaD systems feature cavities, i.e., channels and chambers, through which the liquids move along the assay protocol; the moment of inertia I=I(t) thus increases over time t with the centrifugally outbound motion of (the center of mass of) the liquid distribution Λ(t) .

### 2.4. Critical Spin Rate

The common, normally closed valving schemes of LoaD systems can be described by a first retention phase when the outer meniscus of Λ at r is retained upstream of a critical point in Γ. Subsequently, the rotationally induced release of (part of) the liquid volume $U_{0}$ to an outlet is prompted through shifting the spin protocol ω(t) across a critical frequency Ω. For spin-rate independent axial pressure heads $p_{\to }$ and $p_{\leftarrow }$, and thus also $p_{z}$, we derive a critical frequency
(11)$$\Omega =\sqrt{\frac{p_{\leftarrow }-p_{\to }}{\varrho \cdot \bar{r}\Delta r}}=\sqrt{\frac{p_{z}}{\varrho \cdot \bar{r}\Delta r}}$$
by combining (2) and (6). High- and low-pass valves distinguish by whether the release is triggered by lifting or lowering ω(t) across Ω (11), respectively.

As an alternative to modulating ω(t), the hydrostatic equilibrium (6) may also be altered by adding a liquid volume $V_{\Delta }$ to the inner end of Λ for elevating Δr (and possibly also $\bar{r}$) in $p_{\omega }$ (2). Furthermore, in centrifugo-pneumatic valves delineated by $p_{\to }=p_{0}$ and $p_{\leftarrow }=p_{V}$ (8), the outer compression chamber may be enlarged or vented, i.e., $p_{\leftarrow }=p_{V}\mapsto p_{0}$, for prompting liquid release.

Note that for highly dynamic systems, inertial effects will have to be included in calculating the frequency Ω (11), e.g., by introducing an inertial pressure term $p_{m}>0$ (10) in $p_{\to }$, which would thus lower the effective retention rate Ω compared to the quasi-hydrostatic case mainly considered in the course of this work.

### 2.5. Siphon Valving

From the early days of LoaD systems, siphoning principles have been utilized for centrifugal flow control. In these geometries, Γ, a (usually larger) inlet reservoir is connected by an outlet channel featuring inbound and outbound segments with an inner crest point at $R_{\mathrm{crest}}$ to a receiving chamber. During retention, the front meniscus at r resides in the inbound segment where the centrifugal pressure $p_{\omega }$ (2) points antiparallel to the direction of $p_{z}$, thus effectively stabilizing the retention of the distribution Λ(Ω) against perturbations, such as inertial effects or droplet break-off. This phase is characterized by a Δr between the front meniscus in the inbound section r and the liquid level in the inlet reservoir at $r_{0}$, so that $p_{z}(r)=p_{\omega }\propto \bar{r}\Delta r$ (3). The axial direction of $p_{z}$ determines the sign of Δr during retention of Λ, i.e., whether r is elevated or depressed with respect to $r_{0}$.

In many cases, siphons connect two vented (or very large) chambers, thus neutralizing the impact of the ambient pressure $p_{0}$ applying at either end. In macroscopic setups, priming of the siphon channel is then triggered by the addition of a sufficient volume $V_{\Delta }$ to the inlet for the liquid distribution Λ to progress (sufficiently) past the crest point at $R_{\mathrm{crest}}$. While such release mechanisms can also be deployed for miniaturized systems, LoaD technologies have mainly resorted to priming by capillary action, i.e., $p_{\to }=p_{\Theta }\propto \cos\Theta$ (9), energized by a hydrophilically coated siphon outlet with Θ≪90°. Furthermore, pneumatic pumping through compression of a volume in a side chamber connected to the inlet has been explored [20,28]. In both of these low-pass constructs, either an overflow mechanism, or a continuous liquid “pulley” effect [95] sets in once the condition Δr>0 is met between the menisci in the inner reservoir and the outbound segment; at this stage, also $p_{\omega }$ (2) works in tandem with $p_{z}>0$ to drag the liquid radially outwards, thus further increasing Δr in a self-amplifying pumping mechanism.

In centrifugo-pneumatic (CP) siphon valving (Figure 1), $p_{\to }=p_{0}$ and $p_{\leftarrow }=p_{V}$ (8), i.e., $p_{z}=p_{0}\cdot (V_{0}/V-1)$, so that Δr>0 is needed to drive the liquid past the inner crest point at $R_{\mathrm{crest}}$ all the way to the outlet chamber against shrinking compression volume V, and thus growing $p_{V}\propto 1/V$ (8). Instead of a self-propelling pulley effect promoted by a narrow outlet, advancement of the outbound meniscus requires a steady increase of the spin rate ω(t), and, as long as there is a positive Δr>0, the centrifugal pressure $p_{\omega }\propto \Delta r\cdot \;\omega^{2}$ (2) can, at least mathematically, match any counteracting $p_{V}$ (8).

For a sufficiently narrow outlet channel, the integrity of the liquid plug might be preserved by the surface tension σ along the entire path until the front meniscus at r reaches the entrance of the outer chamber at the radial position $R_{\mathrm{cham}}$; thereafter, the liquid is transferred. However, for larger cross-sections A of the outbound segment of the siphon and high spin rates, the liquid will creep or drip after passing the crest segment $R_{\mathrm{crest}}$ into the outer chamber until Δr=0, rather than showing a quasi-solid, piston-like behavior. There are thus critical frequencies Ω and $\Omega^{*}$ (11) characterizing the retention and release into the outer chamber, which depend on whether the liquid plug is cut at $R_{\mathrm{crest}}$ or $R_{\mathrm{cham}}$. In either case, a volume $U_{\mathrm{loss}}$ remains in the upstream part of the valve.

Note that the critical frequency Ω (11) of CP valves, i.e., their basic as well as their siphon variants, can be widely configured by the original compression volume $V_{C}$; its permanently gas-filled parts can be located at “anywhere”, e.g., where disc real estate is still available, if the fractional volume of its sub-compartments remains pneumatically interconnected.

### 2.6. Centrifugo-Pneumatic Siphon Valving with Dissolvable-Film Membranes

In order to supersede the prerequisite for external actuation to induce venting, CP valves have integrated sacrificial, fluidically impermeable [51] membranes in their final receiving chamber (Figure 1); these seal the compression chamber until they come into contact with their specific solvent, usually an aqueous bioliquid. The opening of these dissolvable films (DFs) requires the transfer of sufficient liquid $U_{\mathrm{DF}}$ to raise the filling level above its location.

## 3. Performance Metrics

In this section, quantitative characteristics for rotationally controlled valving are specified to play a pivotal role in fluidic LSI requiring high packing density that is tightly interwoven with system-level reliability. These metrics are run along with the example of CP-DF siphon valving (Figure 1), but can also be readily employed for further valve types.

### 3.1. High Field Strengths for LUOs

LUOs, such as plasma separation, become most efficient at high field strengths $f_{\omega }$ (1). Given a medium of (differential) density ϱ retained by a valve of critical spin rate Ω (11) at a given position R, we obtain
(12)$${\hat{f}}_{\omega }=\varrho \cdot R\cdot \Omega^{2}$$
for the maximum field strength available for the LUO. As shifting R≈R in (12) mostly impacts $\bar{r}$ in the radial product $\bar{r}\Delta r$ (11), the primary parameter towards maximizing ${\hat{f}}_{\omega }$ (12) for a given retention rate $\omega \leq \Omega \propto 1/\sqrt{\bar{r}\Delta r}$ (11) is Δr, i.e., the liquid distribution Λ(Ω) extending between $r_{0}$ and r, which can be trimmed by the geometry Γ for a given $U_{0}$. The dimensionless metric
(13)$${\bar{f}}_{\omega }=\frac{R}{R_{\max}}\cdot \frac{\Omega^{2}}{\omega_{\max}^{2}}$$
references ${\hat{f}}_{\omega }$ (12) to the largest available field strength at the maximum radial position $R_{\max}$ and (safe) rotational frequency $\omega_{\max}$ that can be produced by the spindle motor.

### 3.2. Radial Space

Due to the unidirectional nature of the centrifugal field $f_{\omega }$ (1), (the center of gravity of) a liquid distribution Λ must always advance in the radially outbound direction. At least in the absence of externally powered centripetal pumping, the processing chambers of subsequent LUOs must thus be staggered radially.

As part of such serial arrangement, the outwards pointing field $f_{\omega }$ (1) puts a price tag on the radial position
(14)$$\bar{R}=\frac{R-R_{\min}}{R_{\max}-R_{\min}}$$
echoing that a central placement is most favorable, i.e., $\bar{R}$ (14) minimized, to keep sufficient room for downstream LUOs. Note that small $\bar{R}$ (14) might interfere with requirements on maximizing the local field strength ${\bar{f}}_{\omega }$ (13) for LUOs. Further, the overall consumption of radial space
(15)$$\bar{\Delta R}=\frac{\hat{r}-\overset{ˇ}{r}}{R_{\max}-R_{\min}}$$
with the inner and outer boundaries $\overset{ˇ}{r}$ and $\hat{r}$ of the structure Γ, respectively, should be taxed towards an optimization of the packing density.

### 3.3. Spatial Footprint

The real estate available for patterning Γ on a discoid carrier is limited, e.g., between $R_{\min}=7.5\;\mathrm{mm}$ outside the 15 mm central hole and $R_{\max}=5.5\;\mathrm{cm}$ to leave an outer rim for bonding a lid, e.g., with a radial extension of 5 mm, for chip geometries derived from conventional optical data storage formats. This makes central real estate, where space is the smallest and most precious.

If we consider a function of the full (azimuthal) width W(r) of Γ along the radial direction r, a simple measure of the spatial footprint would be the entire surface area enclosed by its outer contours confined by $\overset{ˇ}{r}$ and $\hat{r}$. Considering the scarcity of central space, we introduce the metric
(16)$$\bar{A}=\frac{1}{R_{\max}-R_{\min}}\int_{\overset{ˇ}{r}}^{\hat{r}}\frac{W(r)}{2\pi \cdot r}dr$$
which relates the spatial footprint of Γ at R with respect to the total area 2πr⋅dr contained within the ring of (infinitesimal) radial extension dr at a radial location r.

### 3.4. Volume Definition and Loss

Liquid transfer through an LUO and its connected downstream valve may be associated with a volume $U_{\mathrm{loss}}$ left behind in the structure Γ after completion. Depending on the fluidic process of emptying a CP-DF siphon valve, the lost amount of liquid
(17)$$0\leq U_{\mathrm{loss}}\leq (A_{0}+A)\cdot (R-R_{\mathrm{crest}})+U_{\mathrm{iso}}$$
might vanish for a perfect “pulley” mechanism, which maintains a continuous liquid column until the entire liquid in Γ is dragged through the crest channel, and in a pure “overspill” mode until $r=R_{\mathrm{crest}}\;\mathrm{and}$ Δr=0 (Figure 1d). For serial processing, the volume arriving at the next valve $U_{0}^{\prime }=U_{0}-U_{\mathrm{loss}}$ can be factored into its retention rate $\Omega^{\prime }=\Omega (U_{0}^{\prime })$, and also into the mixing ratio of a quantitative bioassay.

The metric
(18)$$\bar{U_{\mathrm{loss}}}=\frac{U_{\mathrm{loss}}}{U_{0}}$$
is useful to quantify volume loss with respect to the loaded volume $U_{0}$. For siphoning, where a significant fraction of the liquid remains in Γ, $\bar{U_{\mathrm{loss}}}$ may be minimized through reduction of $U_{\mathrm{iso}}$ and A⋅Z. As any uncertainty $\Delta U_{\mathrm{loss}}$ directly impacts the spread of the retention rate $\Delta \Omega \text{′}=\Delta \Omega (\Delta U_{\mathrm{loss}})$, minimizing the relative fluctuation
(19)$$\bar{\Delta U_{\mathrm{loss}}}=\frac{\Delta U_{\mathrm{loss}}}{U_{\mathrm{loss}}}$$
is critical for assuring reliability towards fluidic LSI.

### 3.5. Reliability and Band Width

In practice, all experimental input parameters $\left\{\gamma_{k}\right\}$ impacting the liquid distribution $\Lambda =\Lambda (R,\Gamma ,U_{0},\omega )$, and thus Ω (11), are subject to random spreads of standard deviations $\left\{\Delta \gamma_{k}\right\}$ delineated by a Gaussian distribution. This statistical spread implies that at a given spin speed ω, the actual (radial) position of the meniscus r′(ω) will statistically vary around its target value r(ω). So, to make sure that the conditions for operationally reliable flow control are guaranteed, e.g., $r>R_{\mathrm{crest}}$ during retention, the spin rate ω has to stay outside the corridor Ω−M⋅ΔΩ<ω<Ω+M⋅ΔΩ; in other words, ω has to be lifted or lowered through this entire frequency band Ω±M⋅ΔΩ for assuring proper retention and release for high- and low-pass valves, respectively.

The factor M∈{1,2,3,4} represents the degree of (component-level) reliability, with probabilities $P_{M}=\mathrm{erf}(M/\sqrt{2})\approx \{68\%,\;95\%,\;99.7\%,\;99.99\%,\ldots \}$ (and “erf” representing the error function) that the actual values  Ω′ are found in this frequency interval. Above M=6 and M=7, reliability of this (single) valving step is in the range of 10 and 1 defects per million opportunities (DPMO); for M=8, faults are practically absent.

At least in case that the input parameters $\left\{\gamma_{k}\right\}$ are (widely) independent and their spreads $\left\{\Delta \gamma_{k}\right\}$ comparatively small, Gaussian error propagation provides an analytical formula to calculate the standard deviation
(20)$$\Delta \Omega (\left\{\gamma_{k}\},\{\Delta \gamma_{k}\right\})\approx \sqrt{\underset{k}{\sum }{\left(\frac{\partial \Omega }{\partial \gamma_{k}}\cdot \Delta \gamma_{k}\right)}^{2}}$$
of Ω (11), which is a very suitable method for optimization routines. Figure 2 reveals the scaling of Ω and its standard deviation ΔΩ with three impact parameters defined in Figure 1. For a given Γ, the volume of the main compression compartment $V_{C,0}$ (a) and the radial position R (b) are varied; in (c), the field strength $f_{\omega }$. (1) is kept constant while R shifted, e.g., to mimic translocating a particle sedimentation structure requiring a minimum $f_{\omega }$ to a radial position R where space is still available in a multiplexed layout.

Alternatively, Monte Carlo (MC) methods can be interpreted as virtual manufacturing and testing by running a large number of iterations, where a set of values $\left\{\gamma_{k}^{\prime }\right\}$ is obtained from a random number generator assuming Gaussian distributions of $\left\{\gamma_{k}^{\prime }\right\}$ of standard deviations $\left\{\Delta \gamma_{k}\right\}$ around their nominal values $\left\{\gamma_{k}\right\}$ and inserted into Equation (11) for obtaining a histogram for Ω′ in the simulated experiment.

The resulting band widths 2⋅M⋅ΔΩ (20), which are associated with concurrently loaded valves, and thus change over the course of an LUO sequence, need to “fit” into the practically available envelope of spin rates ω between $\omega_{\min}$ and$\;\omega_{\max}$; these boundaries are, on the lower end at $\omega_{\min}$, constrained by a minimum centrifugal field $f_{\omega }$ (1) or pressure $p_{\omega }$ (2) required to stabilize a liquid distribution Λ, e.g., to suppress unwanted capillary motion (9); the upper value at $\omega_{\max}$ is imposed by factors such as the limited torque of the spindle motor $\tau_{\mathrm{spindle}}$ (10), the pressure tightness of the (bonded) lid, and operational safety.

Towards multiplexing, the metric
(21)$$\bar{\Delta \Omega }=\frac{\Delta \Omega }{\omega_{\max}-\omega_{\min}}$$
rewards minimum use of band width 2⋅M⋅ΔΩ (20) with respect to the available frequency envelope.

## 4. Multiplexing

### 4.1. System-Level Robustness

Towards microfluidic LSI, many such serial and parallelized valving steps need to be coordinated on the LoaD device; each component failure might affect the overall reliability. For N independently operating, concurrently loaded valves {i,j} in a given step i, the probability for consistent system-level function amounts to ${\left(P_{M}\right)}^{N}$; for the example of M=2, and thus $P_{M}=95\%$, ${\left(P_{M=2}\right)}^{N}$ drops from approximately 90% to 86%, 77% and 60% for N={2,3,5,10}, respectively, in case the bands $\Omega_{i,j}-M\cdot {\Delta \Omega }_{i,j}<\omega <\Omega_{i,j}+M\cdot {\Delta \Omega }_{i,j}$ must be avoided for properly controlling each valve {i,j}. For N=5 loaded valves, extending the protected frequency intervals from M=2 to M=3 would significantly improve the (fluidic) reliability at the system level from ${\left(P_{M=2}\right)}^{5}=77\%$ to ${\left(P_{M=3}\right)}^{5}=98.5\%$.

Note that the above considerations apply to each step i separately, i.e., after opening a particular valve of the cohort {i,j} at a point in time $T_{i}$, a new group of loaded valves emerges; this fresh set consists of the valves that remained closed in step i, plus the ones next in line to receive the released liquid, excluding “unvalved”, e.g., end-point chambers. The protected areas $\left\{\Omega_{i,j}\pm M\cdot {\Delta \Omega }_{i,j}\right\}$ in the frequency domain thus change over the course of the centrifugally multiplexed assay protocol ω(t).

### 4.2. Frequency Corridor

We consider serial valving in steps {i} with critical spin rates $\left\{\Omega_{i}\right\}$ that are activated at points in time $\left\{T_{i}\right\}$; each $\Omega_{i}$ controls a subset of concurrently loaded and opened valves {i,j}. Retention is assured while the spin rate ω(t) does neither surpass nor drop below the (aggregate) band widths $2\cdot M\cdot \underset{j}{\max}\{{\Delta \Omega }_{i,j}\}$ centered at the critical frequencies $\left\{{\hat{\Omega }}_{i}\right\}$ and $\left\{{\overset{ˇ}{\Omega }}_{i}\right\}$ where $t<T_{i}$ for high- and low-pass valves, respectively. Figure 3a displays the scenario in frequency space for rotational actuation. An initial retention corridor between ${\overset{ˇ}{\Omega }}_{1}$ and ${\hat{\Omega }}_{2}$ successively broadens as further valves are triggered, with ${\hat{\Omega }}_{i}<{\hat{\Omega }}_{i+1}$ and ${\overset{ˇ}{\Omega }}_{i}>{\overset{ˇ}{\Omega }}_{i+1}$.

For the case of actuating (high-pass) CP valves by venting at $\left\{T_{i}\right\}$ with i∈{2,4,5} in the exemplary scenario portrayed in Figure 3b, the retention zone for the spin rate curve ω(t) is also defined by the innermost limits of the allowed bands $\left\{\Omega_{i,j}\pm M\cdot {\Delta \Omega }_{i,j}\right\}$ of the valves still loaded at a given point in time $T_{i}.$ Nevertheless, the order of release is merely imposed by the venting sequence, and does hence not necessitate steadily rising or decreasing critical retention rates ${\hat{\Omega }}_{i}$ and ${\overset{ˇ}{\Omega }}_{i}$ at $T_{i}$, respectively.

### 4.3. Configuration in Real and Frequency Space

Theoretically, the geometry Γ can always be modified to minimize the radial product $\bar{r}\Delta r$ (3), thus allowing any randomly large retention rate $\Omega \propto 1/\sqrt{\bar{r}\Delta r}\;$(11) for a given radial position R and liquid volume $U_{0}$. Yet, its standard deviation ΔΩ (20) might become too broad to still be able to sufficiently multiplex flow control within the confinement of the practical spin rate envelope between $\omega_{\min}$ and $\omega_{\max}$ (Figure 2 and Figure 3). Furthermore, there would be additional restrictions to parameter spaces, e.g., on the smallest feature sizes and tolerances $\left\{\Delta \gamma_{k}\right\}$ (20) associated with the manufacturing technology, or minimum liquid volumes, for instance, as prescribed by the loading and metering technique, the bioassay protocol, or the detection technology.

It also needs to be considered that, especially towards concentrated packing of LoaD substrates, it is wise to leave the liquid carrying parts of geometries Γ, that may have already been thoroughly validated in terms of design-for-manufacture (DfM), fluidic and assay performance, untouched. For CP valving techniques, it is therefore advantageous to only alter larger, uncritical features, e.g., the main volume $V_{C,0}$ of the permanently gas-filled part of the compression chamber (Figure 2a), for tuning Ω (11). Different $U_{0}$ may simply be accommodated by adjusting the width $w_{0}$ of the inlet reservoir (Figure 1). Nonetheless, the paucity of central disc space may necessitate shifting the radial location R, while the retention rate Ω (11) still needs to comply with the planned valving sequence (Figure 4).

Figure 4 shows distinctive scenarios for CP-DF siphon valves commonly encountered when seeking the best compromise in real and frequency space. We again investigate simultaneous actuation of valves {i,j} at times {$T_{i}$}, and sequential firing with isoradial as well as radially inbound and outbound staggering on the given disc. The ordering of $\Omega_{i}$ (11) is accomplished by adjusting the main compression volume $V_{C,0}$, while leaving all other parts of Γ and $U_{0}$ untouched. Note, again, that without non-rotationally powered pumping mechanisms, (the center of gravity of) a given liquid distribution $\Lambda_{i}$ can only migrate radially outbound, i.e., $R_{i+1}>R_{i}$.

Figure 5 displays the distribution of the frequency bands $\left\{\Omega_{i}\pm M\cdot {\Delta \Omega }_{i}\right\}$ in the different rotational actuation scenarios for simultaneous and sequential release (Figure 4) through adjusting the main volume of the compression chamber $V_{C,0,i,j}$. Note that the requirement to create non-overlapping bands Ω±M⋅ΔΩ for serial opening even leads to a slight increase in $V_{C,0}$ towards the highest spin rate ω (Figure 5b), and a huge widening of frequency bands Ω±M⋅ΔΩ in the radially outbound release sequence (Figure 5d).

### 4.4. Exemplary Bioassay Panel

Figure 6 illustrates the rotational automation of a typical bioassay protocol. Plasma P is separated from a (blood) sample S before mixing with a first, onboard stored liquid reagent L1. This mixture P&L1 is then released to a final detection chamber, where it is successively combined with reagents L2 and L3. The liquid handling is coordinated by high-pass valves opening after their respective bands $\Omega_{i}\pm M\cdot {\Delta \Omega }_{i}$ are crossed, thus creating an ω- corridor with a successively growing upper limit over time t. Intermediate mixing steps are implemented by zig-zag patterns in ω(t). The domains available in real and frequency space are confined by $R_{\min}$ and $R_{\max}$, and $\omega_{\min}$ and $\omega_{\max}$, respectively.

The unidirectional nature of the centrifugal field $f_{\omega }$ (1) requires that the positions of the serial processing of the sample through loading, separation, and detection chambers need to follow $R_{S}<R_{P}<R_{D}$, while the release rates of the high-pass valves need to continuously increase along $\omega_{\min}<\Omega_{S}<\Omega_{P}<\Omega_{L1}<\Omega_{L2}<\Omega_{L3}<\omega_{\max}$, so that overlap of their associated bands $\Omega_{i}\pm M\cdot {\Delta \Omega }_{i}$ is avoided. In the portrayed exemplary assay protocol, high quality and speedy plasma separation is linked to a large field strength $f_{\omega }\propto R_{P}\cdot \omega^{2}$ (1), which is limited by $\omega <\Omega_{P}$ (11), thus making an outer position $R_{P}$ most favorable. On the other hand (Figure 2a), high retention rates $\Omega_{P}<\Omega_{L,i}\propto 1/\sqrt{{\bar{r}}_{i}\Delta r_{i}}$ (11) for the onboard reagents within the narrow corridor $\Omega_{P}<\omega <\omega_{\max}$ are best achieved for their more central placement (when estimating ${\bar{r}}_{i}\approx R_{L,i}$ and $\Delta r_{i}\approx \mathrm{const}.$).

Overall, the structures $\left\{\Gamma_{i,j}\right\}$ presenting LUOs with their downstream valves need to meet several, partially contradictory demands on the maximum retention rate Ω (11), disc real estate $\bar{A}$ (16), maximum field strength ${\hat{f}}_{\omega }$ (12), and footprints in radial $\bar{R}$ (14), $\bar{\Delta R}$ (15), and frequency space $\bar{\Delta \Omega }$ (21).

Figure 7 shows the release rates $\left\{\Omega_{i}\right\}$ for given $\Omega_{S}/2\pi =15\;\mathrm{Hz}$ and $\Omega_{P}/2\pi =30\;\mathrm{Hz}$ (at $R_{P}=4\;\mathrm{cm}$) of CP-DF siphon valves with the same default geometries $\Gamma_{i}$, except for their (main) compression volumes $V_{C,0,i}$, which are varied to guarantee disjunct tolerance intervals $\left\{2\cdot M\cdot {\Delta \Omega }_{i}\right\}$ for a given M=3. The two distributions represent sets of identical $\left\{\Omega_{i}\right\}$ with their radial positions $\left\{R_{i}\right\}$ shifted by 0.5 cm as a possible way to increase the functional integration density on a given disc cartridge. Note that the rotational actuation of these high-pass valves “squeezes” the release rates for the onboard reagents $\left\{\Omega_{L,i}\right\}$ in the narrow ω− strip within $\Omega_{P}+M\cdot {\Delta \Omega }_{P}<\omega <\omega_{\max}$ (see also Figure 6b).

### 4.5. Assay Parallelization

A straight-forward method to replicate several simultaneously processed, identical assay protocols on the same LoaD carrier, e.g., for testing different samples for the same analytes, or for redundancies for controls, calibration, or statistics, is to set equal release frequencies $\left\{\Omega_{i}\right\}$, which are crossed synchronously $\left\{T_{i}\right\}$. The packing density might be augmented by the radial staggering of the structures Γ, for which Figure 2 shows the required adjustments of the main compression volumes $V_{C,0,i,j}$, and their impact of the total band width $2\cdot M\cdot {\Delta \Omega }_{i}$. Concurrent processing of homologous assay protocols with the (primary) variation of inlet volumes $U_{0,i}$ may be readily achieved by adjusting the cross-sections $A_{0}$ of the inlet reservoirs (Figure 1). In the general case of diverging multiplexing protocols, each release step must comply with the frequency corridor required to reliably shut the concurrently loaded valves.

## 5. Advanced Rotational Flow Control

We now expand on the actuation mode of CP-DF siphon valves by venting (Figure 1d). Instead of the various mechanical, radiation-induced, or thermal methods to open the compression chamber, a second (ancillary) liquid volume wets a DF placed at a remote location to effectuate $p_{V}\mapsto p_{0}$. First, a handshake mechanism called “event triggering” is introduced, which also enables fluidic conditionals logical flow control. Afterward, the timing of event triggering, e.g., to comply with mixing or incubation periods of bioassays, through delay elements and rotational pulsing, is outlined.

### 5.1. Event Triggering

Two DFs are attached to the compression chamber of CP-DF siphon valves (Figure 8). In an exemplary assembly of upper and lower layers displaying the microfluidic conduits sandwiching a sealing sheet containing a set of DFs, the first “Load Film” (LF) initially blocks the outlet for the liquid through a vertical via into a lower layer; a second “Control Film” (CF) is placed in a pneumatically connected, distal compartment [81]. Below a critical spin rate Ω (11), the pneumatic counterpressure $p_{V}$ (8) prevents the liquid from reaching the LF.

The remote CF can be accessed by another ancillary liquid through a channel, e.g., situated in the lower layer, to open the DF, to thus massively expand the enclosed gas volume $V_{C}\mapsto \infty$. With $p_{V}\mapsto p_{0}$, the main liquid advances into the chamber containing the LF, which then disintegrates for release through the exit channel.

Event triggering (Figure 8) refers to the constellation when a liquid received from a preceding stage of an assay protocol takes on the role of the ancillary liquid to establish a “handshake” type flow control; instead of rotational cues or external power, the sequential release follows a “domino effect” that proceeds broadly independent of the spin rate ω(t). The frequency corridor (Figure 3), and especially the maximum field strength ${\hat{f}}_{\omega }$ (12) that is available for LUOs along the assay procedure, are thus only restricted by the (minimum) retention rate Ω (11) of concurrently loaded (high-pass) valves, or $\omega_{\max}$.

Figure 9 illustrates the benefits of event-triggered flow control regarding configurability and packing density in real and frequency space. As opposed to the implementation of the exemplary bioassays with the rotationally actuated high-pass CP-DF siphon valves (Figure 7), the liquid reagents $\left\{L_{i}\right\}$ may all be released $\omega >\Omega_{P}+M\cdot {\Delta \Omega }_{P}$ with respect to the plasma separation at $\Omega_{P}$. Overall, the speed (and quality) of plasma separation in the initial stages of common assay protocols may hence be significantly improved.

### 5.2. Logical Flow Control

Such an event-triggered relay may also be considered as a logical “IF” condition, in a way that a (subsequent) valve only fires “if” another liquid has arrived to dissolve the CF in a designated pneumatic compartment [81]. This concept has been further refined by placing the CF at critical radial positions $R_{\mathrm{DF}}$ (Figure 1a) in a way that a minimum liquid volume $U_{\mathrm{CF}}$ must arrive in the DF chamber to dissolve the membrane, and thus vent and open the next valve. Based on such a logical “IF” condition, closed-loop “handshake” flow control has been implemented. Alternatively, the DF in the receiving chamber may be placed at a radial position that is only reached after a minimum amount of liquid has entered from (at least) two incoming channels, thus establishing an “AND” element. In a similar way, a logical “OR” may be defined by positioning the DF so that it is wetted after the arrival of a minimum liquid volume from a defined subset of incoming channels.

### 5.3. Delay Elements

Bioassay protocols usually require certain time spans, e.g., to account for reaction kinetics between bioentities that are dissolved, suspended, or bound to surfaces on cells and beads of walls. However, event triggering “hard-wires” process intervals in the network of valving structures in a direct “knock-on” fashion. To this end, delay elements have been incorporated into event-triggered fluidic systems for bioassay integration [96]. Examples are hydrodynamic flow resistances imposed by narrow and long channel segments, or paper strips, which have also been utilized for other purposes [15,97]. Through imbibition at defined wicking speed, these timers slow down the transfer of the main or ancillary liquids compared to open channels. While, in theory, also reducing the spin speed during transfer periods would be an option, such action would require rather complex “closed-loop” coordination between monitoring liquid flow on the disc and the frequency of the spindle motor ω.

### 5.4. Pulsing of Spin Rate

Commonly, the aggregate time spans for dissolution and liquid transfer range on the second- to minute scale, while bioassays normally require processing times of minutes to an hour. So, while such physical delay elements prolong the intervals available for LUOs, the timing still faces upper limits, and it is also ‘inscribed’ in the layout and material characteristics, such as the dissolution time of the sacrificial membrane or the imbibition speed of the paper strip. In addition, the requirement of the rotational actuation mode to steadily increase the critical spin rates $\Omega_{i}$ along successive steps i managed by high-pass CP valves compromises the flexibility that may be required for enabling microfluidic LSI.

For situations when liquid volumes $\left\{U_{i,j}\right\}$ are synchronously transferred between serial steps {i} at $\left\{\Omega_{i}\right\}$ and $\left\{T_{i}\right\}$, the concept of rotational pulsing (Figure 10) has been presented [82]. This flow control mechanism exploits that within the interval $\Delta T_{i}=T_{i+1}-T_{i}$, the subsequent valve i+1 is not fully loaded with $U_{0}$, yet, thus intermittently increasing is retention rate $\Omega_{i+1}$ (Figure 10c) to allow jumps in the spin rate ω(t) that only need to stay inside the corridor required to retain the (other) concurrently loaded liquids; furthermore, the spin rate ω may be reduced below $\Omega_{i}$, thus alleviating the requirement for a steady increase (Figure 7) or constant spin rate ω (Figure 8 and Figure 9) while sequentially progressing the release steps {i}.

In a similar way as ‘venting’, rotational pulsing allows retaining liquid menisci at sufficiently “safe” spacing at r=r(Ω−M⋅ΔΩ) from the DF during retention. It can also be used to synchronize parallel processes, e.g., when the same protocol needs to be run on several patient samples. However, in heterogeneous assay panels where each channel would run a different protocol, the height of the rotational pulse for the CP-DF valve i to be released is confined by the lowest critical frequency of the concurrently loaded valves {κ} that need to stay closed, i.e., $\omega_{\mathrm{pulse},i}<\underset{\kappa \neq i\;}{\min}\{\Omega_{\kappa }-M\cdot \Delta M_{\kappa }\}$.

## 6. Conclusions and Outlook

“Lab-on-a-Disc” (LoaD) systems are highly promising candidates for enabling cost-efficient and robust commercial solutions for decentralized testing, especially in low-resource settings and minimally trained operators. Underlying sample-to-answer automation and parallelization of bioassay protocols roots in fluidic large(r)-scale integration (LSI) of Laboratory Unit Operations (LUOs) and their control valves at tight packing densities.

This work has elaborated the basic principles of centrifugal flow control to directly connect experimental parameters such as liquid volumes, geometrical dimensions, and the ambient atmospheric pressure to the retention and release rates of rotationally actuated valves at the pivot of LSI. In the wake of such “digital-twin” modeling, frequency bands are calculated from tolerances in the input variables to quantify operational robustness. As the practically limited real and frequency spaces of LoaD systems severely restrict fluidic LSI, a suite of metrics has been elaborated to algorithmically guide performance optimization. Furthermore, the principles of advanced rotational valving schemes enabling conditional, closed-loop flow control and random timing by “digital” pulsing were systematically explored. With the availability of extended fabrication capabilities that are beyond the reach of most academic laboratories, larger numbers of LoaD devices can be provided for statistically relevant validation and refinement of the digital-twin model.

Future work will expand the design tool to allow for the simulation of mixed layouts synergistically combining various types of high- and low-pass valves, for refined geometries exhibiting rounded edges and contours, for incorporating dynamic fluid effects and bioassay kinetics, for onboard storage and release of reagents, for bead-based multiplexing techniques [24], and for externally powered, onboard pumping and control modules [76]. The degree of multiplexing might be markedly enhanced by multi-disc stack architectures.

Furthermore, manufacturing processes may be virtualized, e.g., through mold flow analysis for commercial polymer replication techniques, which commonly indicates the need for fairly even distributions of fluidic structures and flow resistances (for the centrally injected hot melt), minimum draft angles, and surface roughness to facilitate proper demolding. Design guidelines are also proving to be important for assembly and bonding.

With its high predictive power, the presented work and the suggested follow-ups can enable the computational generation and validation of disc layouts regarding fluidic functionality and compliance with design-for-manufacture (DfM) guidelines and related scale-up of mass fabrication. Such software might then create an effective interface for foundry services as a hallmark of mature industrial supply chains in task-sharing economies. Superior control and knowledge of manufacturing tolerances can also empower anti-counterfeit strategies protecting genuine medical products and their users [98].

Most foundational patents on centrifugal microfluidics that were primarily filed throughout the 1990s and early 2000s have expired, thus creating substantial wiggle room in the intellectual property (IP) space. Apart from classical exploitation, such opportunity might also resonate well with the present highly disruptive socio-economic and technological trends in the technological ecosystem of LoaD, such as Maker/Fab Labs, hackathons, science as a service (SaaS), cloud computing, Machine Learning (ML)/Artificial Intelligence (AI), Big Data, Internet of Things (IoT), and decentralization. Overall, this development may enable novel open platform concepts, e.g., to usher a new era of blockchain-backed participatory research models and crowdsourcing of ideas, expertise, skills, workforce, infrastructure, and equipment may be established [99,100,101,102].

## Figures and Tables

**Figure 1 micromachines-12-00700-f001:**
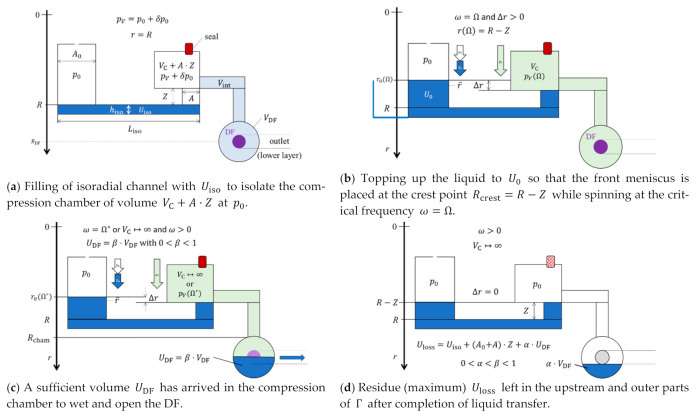
CP-DF siphon valving. The measures of the default geometry Γ (linearized display, not to scale) are compiled in Appendix A, Table A1. The structure consists of an inlet reservoir of cross-section $A_{0}=d_{0}\cdot w_{0}$ with depth $w_{0}$ and width $w_{0}$, which is connected through a (narrow) isoradial channel of volume $U_{\mathrm{iso}}=d_{\mathrm{iso}}\cdot L_{\mathrm{iso}}\cdot h_{\mathrm{iso}}$ positioned at R to an inbound section of radial extension Z and cross-section A=d⋅w of depth d and width w. (For the sake of simplicity, $d_{0}=d_{\mathrm{iso}}=d=1\;\mathrm{mm}$ is chosen in this rudimentary layout.) During retention, the outer part represents the gas-filled compression chamber, with a main compartment of volume $V_{C,0}$, isoradial and radial sections of aggregate volume $V_{\mathrm{int}}$ (with $V_{\mathrm{int}}\ll V_{C,0}$) connecting to a shallow round chamber of volume $V_{\mathrm{DF}}$ (with $V_{\mathrm{DF}}\ll V_{C,0}$) centered at $R_{\mathrm{DF}}$. This outer compartment of volume $V_{\mathrm{DF}}$ features a dissolvable-film (DF) membrane. This DF intermittently covers a centrally placed vertical via leading to an outlet in a lower layer until a liquid volume $U_{\mathrm{DF}}=\beta \cdot V_{\mathrm{DF}}$ with geometry-dependent coefficient 0<β<1 (here: α=1/2) has arrived. A fraction $\alpha \cdot V_{\mathrm{DF}}$ with α<β remains in the receiving chamber after transfer.

**Figure 2 micromachines-12-00700-f002:**
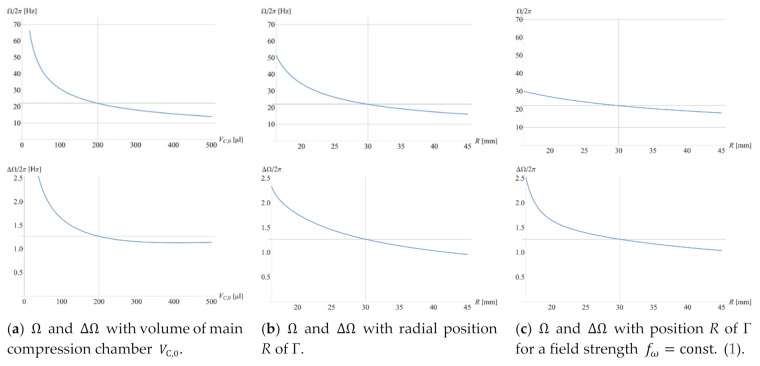
Variation of retention rate Ω (11) and its standard deviation ΔΩ (20) with (**a**) the volume of the main, permanently gas-filled compartment of the compression chamber $V_{C,0}$ and (**b**) the radial position R while leaving (the remainder of) the structure Γ unchanged. The retention rate Ω sharply increases towards shrinking compression volumes $V_{C,0}$ and central placements R; this comes at the expense of widening the tolerance ΔΩ. The gridlines represent the default volume $V_{C,0}=200\;µL$, at a radial position R=30 mm and retention rate Ω/2π≈22 Hz. (**c**) Ω and ΔΩ vs. R while maintaining the field strength $f_{\omega }$ (1), originally evaluated for its default geometry Γ (Table A1), critical spin rate Ω/2π=30 Hz, $\varrho =1000\;{\mathrm{kg}\;m}^{-3}\;$ and R=3 cm, while varying the radial position R.

**Figure 3 micromachines-12-00700-f003:**
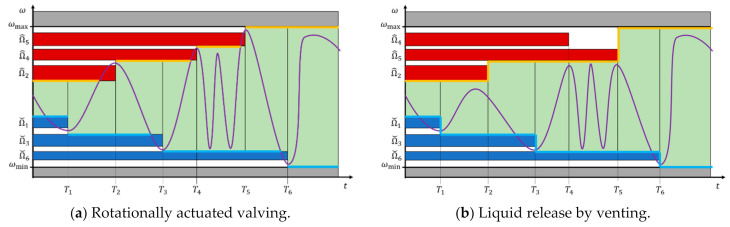
Multiplexing of concurrently loaded valves in frequency vs. time domain. High- and low-pass valving takes place at points in time $\left\{T_{i}\right\}$. With each step i, a certain frequency corridor (green) that is available for LUOs rearranges. (**a**) For rotational actuation, reliable opening of valves {i,j} in a given step i is triggered by fully crossing their linked frequency bands $\left\{\Omega_{i,j}\pm M\cdot {\Delta \Omega }_{i,j}\right\}$. (**b**) In venting mode, the concurrently loaded valves serially release according to the order of opening their compression chamber, i.e., $V_{C,0,i}\mapsto \infty$ and $p_{V}\mapsto p_{0}$.

**Figure 4 micromachines-12-00700-f004:**
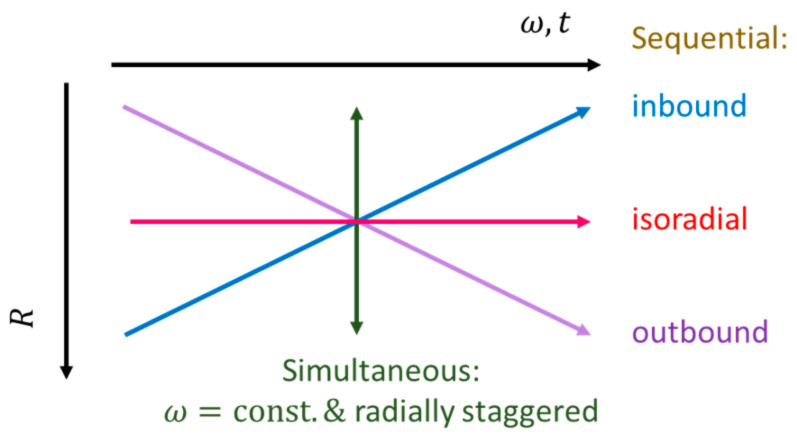
Representative scenarios underpinning multiplexing. The critical retention rates $\left\{\Omega_{i,j}\right\}$ (11) of concurrently loaded (high-pass) valves {i,j} to be rotationally actuated at a time $T_{i}$ may be preserved while shifting the radial positions $\left\{R_{i,j}\right\}$ of the structures $\left\{\Gamma_{i,j}\right\}$. This might be, for instance, required to meet spatial needs towards more crowded disc layouts, by adjusting the (permanently gas-filled part) of the compression chamber of volume $V_{C,0,i,j}$, while observing proper arrangement in the frequency domain 𝜔, e.g., to avoid overlapping bands by $\Omega_{i}+M\cdot {\Delta \Omega }_{i}<\Omega_{i+1}-M\cdot {\Delta \Omega }_{i+1}$.

**Figure 5 micromachines-12-00700-f005:**
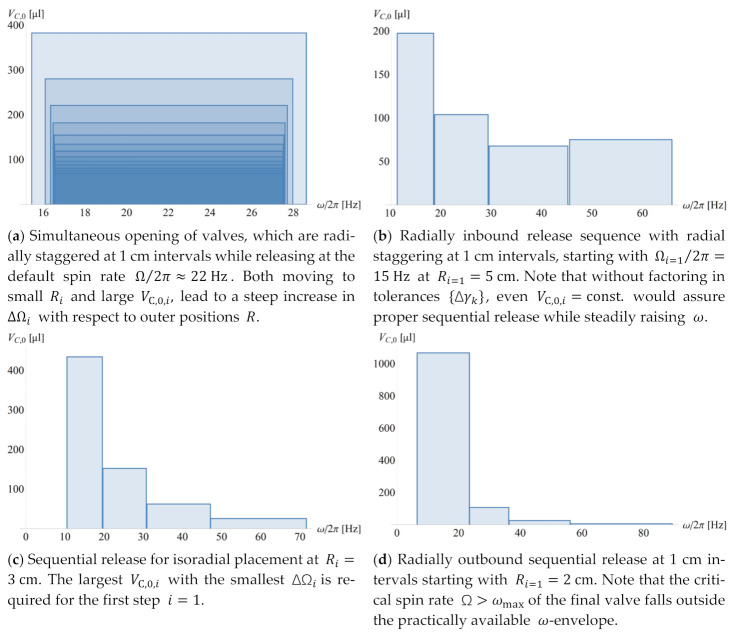
Distribution of bands in frequency space ω with required volumes of the main compression chamber $V_{C,0,i}$ for the different radial staggering and release scenarios as portrayed in Figure 4. CP-DF siphon valves {i,j} to be released simultaneously, i.e., $T_{i}=T_{i+1}$, or sequentially, i.e.,$\;T_{i}<T_{i+1}$; the valves, which have similar geometries $\Gamma_{i,j}$ of their liquid-occupied sectors and downstream compression volumes $V_{C,0,i,j}$, are placed isoradially, i.e., at R=3 cm, or staggered over radial positions R={2,3,4,5} cm. The reliability factor is M=4. (**a**) A set of radially staggered valves {i,j} with $R_{i,j}>R_{i,j-1}$ has been tuned through their $V_{C,0,i,j}$ to simultaneously burst in the same step i once $\omega >\Omega_{i}+M\cdot {\Delta \Omega }_{i}$ with ${\Delta \Omega }_{i}=\max\{{\Delta \Omega }_{i,j}\}$. There are three representative cases for sequential actuation with $\Omega_{i}-M\cdot {\Delta \Omega }_{i}>\Omega_{i-1}+M\cdot {\Delta \Omega }_{i-1}$ for successive steps i according to $\Omega_{i}>\Omega_{i-1}$. (**b**) For identical $\Gamma_{i,j}$ and $V_{C,i,j}$, the valves will open according to their radially inbound order $R_{i}<R_{i-1}$. (**c**) For isoradial alignment $R_{i}=R=\mathrm{const}.$, the valve with the largest $V_{C}$ opens first: $V_{C,i}<V_{C,i-1}$. (**d**) Further, a radially outbound opening sequence $R_{i}>R_{i-1}$ can be achieved, for which $V_{C,i}$ needs to be drastically reduced along successive steps i to suppress premature release from by the high pressure $p_{\omega }$ (2) at distal locations R, which comes at the expense of huge bandwidth ΔΩ (20).

**Figure 6 micromachines-12-00700-f006:**
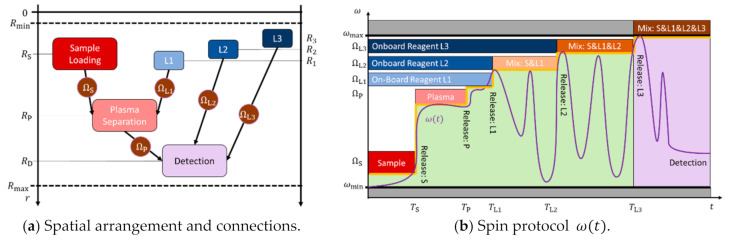
Exemplary bioassay implementing the loading of blood sample, plasma separation, and mixing with three prestored liquid reagents $\left\{L_{i}\right\}$. Flow is controlled by five high-pass valves with release rates $\omega_{\min}<\Omega_{i}<\omega_{\max}$ located at $R_{\min}<R_{i}<R_{\max}$ and opening for $\omega >\Omega_{i}\;$at points in time $T_{i}$. (**a**) Plasma (P) is extracted from the sample (S) in a peripheral position $R_{P}$ at high field strength $f_{\omega }\propto R_{P}\cdot \omega^{2}$ (1). While owing to the unidirectional nature of the centrifugal field $f_{\omega }$ (1), the course of the assay requires $R_{S}<R_{P}<R_{D}$ and $R_{L1}<R_{P}$, a radially inbound staggering $R_{L1}>R_{L2}>R_{L3}$ (Figure 5b) has been chosen for the serial release of $\left\{L_{i}\right\}$ through their high-pass valves opening at $\Omega_{L1}<\Omega_{L2}<\Omega_{L3}$. (**b**) A blood sample S is loaded and released at $\Omega_{S}<\omega <\Omega_{P}$. Separation of plasma P proceeds at $\Omega_{P}<\omega <\Omega_{L1}$. Onboard liquid reagent L1 is then forwarded at $\Omega_{L1}<\omega <\Omega_{L2}$ into the separation chamber where it is mixed with the plasma P within $\omega_{\min}<\omega <\Omega_{L2}$. The mixture P&L1 then progresses to the final detection chamber for $\Omega_{P}<\omega <\Omega_{L2}$, followed by addition of and mixing with L2 within $\Omega_{2}<\omega <\Omega_{3}$ and $\omega_{\min}<\omega <\Omega_{L3}$, respectively. Finally, L3 is added at $\Omega_{3}<\omega <\omega_{\max}$ and blended to obtain S&L1&L2 within $\omega_{\min}<\omega <\omega_{\max}$. The green background marks the allowed frequency corridor at each point in time $t=T_{i}$, with its upper limit expanding according to $\omega_{\min}<\Omega_{S}<\Omega_{P}<\Omega_{L1}<\Omega_{L2}<\Omega_{L3}<\omega_{\max}$. Note that, for the sake of simplicity, the frequency thresholds $\left\{\Omega_{i}\right\}$ are meant to refer to their associated bands $\Omega_{i}\pm M\cdot {\Delta \Omega }_{i}$.

**Figure 7 micromachines-12-00700-f007:**
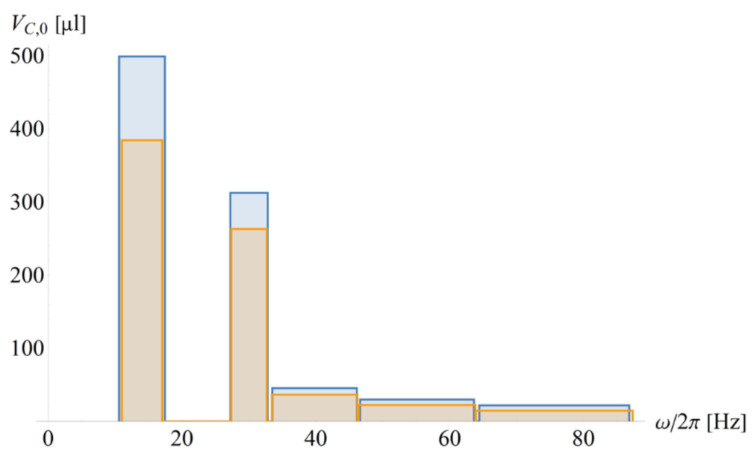
Rotational automation of an exemplary bioassay featuring sequential release of sample and separated plasma as well as pre-stored liquid onboard reagents L1, L2, and L3 with high-pass CP-DF siphon valves of release rates $\left\{\Omega_{i}\right\}$ with $\Omega_{S}/2\pi =15\;\mathrm{Hz}$ and $\Omega_{P}/2\pi =30\;\mathrm{Hz}$. The release rates $\left\{\Omega_{i}\right\}$ are tuned by the dead volumes of the main compression chamber $\left\{V_{C,0,i}\right\}$, as displayed on the vertical axis. (a) Radial positions (blue) $R_{i}$={3,4,3.5,3,2.5} cm and (orange) $R_{i}^{\prime }=R_{i}+0.5\;\mathrm{cm}\;$mimick concurrent processing of the same bioassay protocols requiring identical $\left\{\Omega_{i}\right\}$ for concurrent valving (M=3), while increasing their spatial packing density towards microfluidic LSI.

**Figure 8 micromachines-12-00700-f008:**
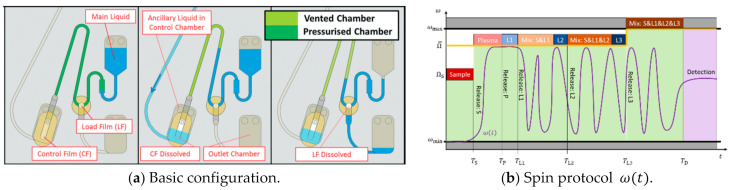
Principle of event-triggered flow control. (**a**) Basic valve configuration with the control film (CF), which is opened by a first liquid to vent the compression chamber of a pneumatic valve. Consequently, a second liquid is released through the load film (LF) (Adopted from [81] with permission from The Royal Society of Chemistry). (**b**) Extended corridor for the spin rate ω(t) in case of the event-triggered versus the rotationally actuated (Figure 6a) high-pass valves. After the sample is released at $\omega >\Omega_{S}$, the upper boundary is defined by the, e.g., common, retention rate $\overset{=}{\Omega }$ shared by all LUOs. Release from their chambers is prompted by event-triggered venting, rather than raising the spin rate ω.

**Figure 9 micromachines-12-00700-f009:**
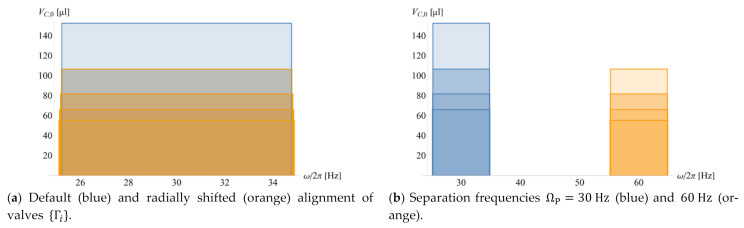
Event-triggered flow control in exemplary bioassay protocol. (**a**) The set of valves $\left\{\Gamma_{i}\right\}$ at their default (blue) positions $\left\{R_{i}\right\}$, and radially shifted outwards by 0.5 cm (orange), which is a common requirement towards microfluidic LSI. All frequency bands $\left\{\Omega_{i}\pm M\cdot {\Delta \Omega }_{i}\right\}$ are centered at identical rates $\Omega_{i}=\Omega$. (**b**) The same radial distribution $\left\{R_{i}\right\}$ is operated at different spin rates $\Omega_{i}=\Omega_{P}$, showing that event triggering is advantageous for enabling high spin rates ω, e.g., for improving and accelerating plasma separation.

**Figure 10 micromachines-12-00700-f010:**
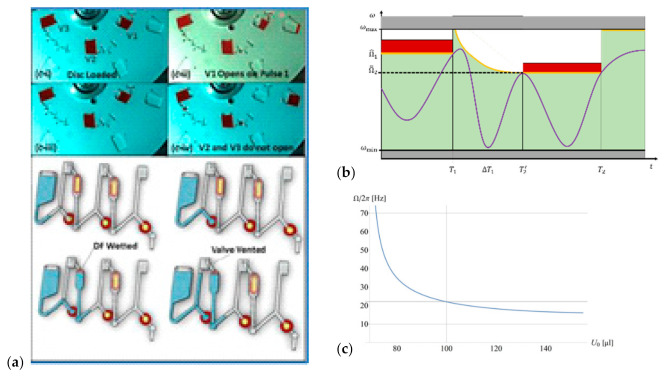
Pulse-actuated CP-DF valving (Adopted from [82]). (**a**) Photo and concept of the fluidic structure. (**b**) Spin protocol ω(t) and (**c**) decrease of the retention frequency $\Omega (U_{0})$ with the arrival of the liquid volume $U_{0}(t)$. In conventional CP-DF siphon valving (Figure 1), a group of liquids is sequentially released through stepping up the spin rate ω across their critical burst frequencies $\left\{\Omega_{i}\right\}$. Digital pulse actuation considers the time $\Delta T_{i}$ it takes a liquid volume $U_{0}$ to advance through the outlet channel between valve opening at the burst frequency $\Omega_{i}$, and its arrival at the following valve featuring $\Omega_{i+1}$. During this interval $\Delta T_{i}$, which is primarily determined by the centrifugally induced pressure head $p_{\omega }$ (2), the flow resistance of the connecting channel on the viscosity of the liquid, and the dissolution time of the DF, the subsequent valve i+1 is only partially loaded, so its effective retention rate $\Omega_{i+1}(T_{i}<t<T_{i+1})$ is transiently be lifted above $\Omega_{i+1}(U_{0})$ that is nominally calculated after the arrival of the full volume $U_{0}$ (**c**). The spin rate ω(t) can thus be spiked well above $\Omega_{i+1}(U_{0})$ during this interval $\Delta T_{i}$, and also allow $\Omega_{i+1}<\Omega_{i}$ to squash the requirement of steadily growing retention rates $\Omega_{i+1}>\Omega_{i}\;$for the high-pass valves along the serial assay protocol ω(t).

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
