# Peer review of "Secure Air Traffic Control at the Hub of Multiplexing on the Centrifugo-Pneumatic Lab-on-a-Disc Platform"

_micromachines, 2021, doi:10.3390/mi12060700_

Round 1

Reviewer 1 Report

The manuscript describes in detail some key parameters for designing centrifugo-pneumatically coupled valve networks for lab-on-a-disc platforms. The manuscript is well written and its contents are appropriate for the readers of the Micromachines journal. I just have three minor points:

1) Please discuss the potential role of fluid viscosity (if any) on the operation of the CP valve shown in this work. 

2) The diagram of CP-DF siphon valving shown in Figure 1 must be improved for clarity as currently it is very difficult to understand. By looking at the current diagram, it is not possible to know that the valve is composed of channels with varying depth (inlet reservoir vs isoradial channel vs inbound section, having depth of d_0, d_iso, and d, respectively).

3) I suggest the author to include a table summarizing all the symbols used in the manuscript.

Author Response

Reviewer 1

Comments and Suggestions for Authors

The manuscript describes in detail some key parameters for designing centrifugo-pneumatically coupled valve networks for lab-on-a-disc platforms. The manuscript is well written and its contents are appropriate for the readers of the Micromachines journal.

Thank you very much for this kind assessment of our manuscript.

I just have three minor points:

1) Please discuss the potential role of fluid viscosity (if any) on the operation of the CP valve shown in this work.

Two short sentences have been added about the role of the viscosity to the text. Due to the essentially hydrostatic nature of the valving principle, flow-based effects involving the viscosity are not of immediate relevance to the core understanding for this paper, but is indeed good to have viscosity mentioned.

2) The diagram of CP-DF siphon valving shown in Figure 1 must be improved for clarity as currently it is very difficult to understand. By looking at the current diagram, it is not possible to know that the valve is composed of channels with varying depth (inlet reservoir vs isoradial channel vs inbound section, having depth of d_0, d_iso, and d, respectively).

Thanks for pointing this out. The caption of figure 1 has been significantly updated for a more thorough introduction of the variables; furthermore, a sentence stating about the identical depths of the main segment of the portrayed structure  has been added.

3) I suggest the author to include a table summarizing all the symbols used in the manuscript.

In combination with the updated Figure 1 and its caption, Table A1 summarizes the definition of the variables.

Reviewer 2 Report

This manuscript describes a virtual “digital twin” strategy that considers experimental tolerances, to efficiently design such “Lab-on-a-Disc” systems. Use this concept, the author focused on gearing centrifugo-pneumatically coupled valving networks which can be utilized for large-scale integration (LSI) of parallelized, multi-step bioassay panels. It is an important resaech covering the analysis and prediction for the centrifugal microfluidic technology. The manuscript is informative and well written. Due to the points summarized above, I conclude that this manuscript needs the following minor revision before it can be resubmitted.

Minor comments:

  1. Fig. 10. The figure legend is not described clearly. Please mark (a) and (b) in the figure.
  2. P17, abbreviations have been defined. No need to repeat the full words.

Author Response

This manuscript describes a virtual “digital twin” strategy that considers experimental tolerances, to efficiently design such “Lab-on-a-Disc” systems. Use this concept, the author focused on gearing centrifugo-pneumatically coupled valving networks which can be utilized for large-scale integration (LSI) of parallelized, multi-step bioassay panels. It is an important research covering the analysis and prediction for the centrifugal microfluidic technology. The manuscript is informative and well written. Due to the points summarized above, I conclude that this manuscript needs the following minor revision before it can be resubmitted.

Thank you for this fair account of the research presented in the manuscript. Much appreciated.

 Minor comments:

  1. Fig. 10. The figure legend is not described clearly. Please mark (a) and (b) in the figure.

The sub-figures have been labelled with a, b and c.

  1. P17, abbreviations have been defined. No need to repeat the full words.

The manuscript was checked and corrected for repeated definition of abbreviations.

Reviewer 3 Report

This is a very original and valuable text that requires further polishing and editing before publication. Although the article has a confusing title, an abstract and introduction that do not explain the content of the article, I still support its publication due to the unorthodox way it discusses the topic. It was a fresh breath of air amongst the literature on LoD systems that has a lot of repetition. The article has 4 sections:  i) Basics of rotational flow control (review of governing principles of driving force and valving), ii) Performance metrics, iii) Multiplexing (the only part that is directly related to the claimed large scale integrated LoD), iv) Advanced Rotational flow control (the siphon valves with vent). Throughout the article it is not possible to differentiate the new information presented in this article and the known literature. The well-known operation principles are explained with some partial referencing. Sections i and ii have very little new information and are more appropriate for a review article. 

If the author is claiming to present the design principles of large-scale integration, the abstract and introduction should clearly state this and also some results should be shown. The title reminds me of the classic Science cover article of VLSI microfluidics of two layer pneumatic valves, Quake group, v 298, 5593, 2002. The emphasis on multiplexing and LSI throughout the article raises the expectation of the reader to see such an example. Instead, the article discusses the operation principles of LoDs and most commonly used valving mechanisms. In its current form, this article is much more suitable as a review article. Some parts of the article (especially introduction and conclusion) reads more like an editorial than a research article. Some buzzwords from other disciplines are used that makes the whole article much more fun to read. However, it does not fit to a research article since it does not give any relevant information. In some cases (such as the title, the digital twin analogy and the last paragraph on AI, Big Data, IoT, crowdsourcing, blockchain, etc..) these analogies are distracting rather than being helpful. I think the last paragraph is completely out of context despite the fact that it is fun to read. I also disagree with the digital twin analogy that was used throughput the article. The principles and guidelines given here are better represented as well-known engineering design principles. The digital twin analogy has a very different connotation than the ideas presented in this article. 

Overall, what is informative is the discussion on multiplexing section. Although, it presents the well-known facts of yield calculation and filter design principles used in micromanufacturing and circuit design, the discussion is helpful for integration of multiple valves and a LoD assay flow design. The spatial and frequency domain representations are very helpful in conveying the author's point (Fig 6). A conceptual assay example is given. In the last section, the hand-shake and triggering analogies explains the principle. Fig. 9 nicely illustrates the benefit of vented channel design. 

To me such a manuscript is much more suitable as a review article with a more focused title, abstract, introduction and conclusion. If the author prefers a research article, the demonstration of these principles as a LSI LoD demonstration is almost a necessity. Then, also the new information in this article should be clearly differentiated than the existing literature, preferably using a more technical and dull language than the author prefers. 

Thank you for this very original article and I hope you take my comments as constructively and find them useful in revising the work. I recommend a major revision of the article mostly due to the structural comments I had above. As a reader I would personally enjoy such a text very much as a review article. 

Author Response

This is a very original and valuable text that requires further polishing and editing before publication. Although the article has a confusing title, an abstract and introduction that do not explain the content of the article, I still support its publication due to the unorthodox way it discusses the topic. It was a fresh breath of air amongst the literature on LoD systems that has a lot of repetition.

Your positive feedback is very welcome. I felt that in addition to the often excellent, but, at times, indeed repetitive work published in the field, a conceptual modelling paper on centrifugal microfluidics might present a value to the community. Especially for the application case of centrifugo-pneumatic flow control, the - often significant - impact of tolerances, e.g., machining, pipetting / metering and ambient pressure, would be worthwhile covering in a systematic fashion by the digital twin strategy.

The article has 4 sections:  i) Basics of rotational flow control (review of governing principles of driving force and valving), ii) Performance metrics, iii) Multiplexing (the only part that is directly related to the claimed large scale integrated LoD), iv) Advanced Rotational flow control (the siphon valves with vent). Throughout the article it is not possible to differentiate the new information presented in this article and the known literature. The well-known operation principles are explained with some partial referencing. Sections i and ii have very little new information and are more appropriate for a review article.

The reviewer addresses the issue whether to classify the work as original research and or review type. By the nature of the outlined “digital twin” approach, and to avoid losing the many readers who might not be so familiar with the topic straight from the start, the work presented indeed set the foundations of the model-based approach, which is also important to define the key parameters. While the reviewer is correct in that the valving mechanisms themselves have been introduced before, the novelty of this manuscript lies in their unprecedented and unified modelling approach for guiding the disc layout, and the decisive impact of (unavoidable) tolerances in experimental input parameters on reliability.
Further references were added to the manuscript.
Regarding the title, it is of course a matter of preference whether to present a more technical or more inspiring title. The idea leading phrasing of the title is that the centrifugo-pneumatic flow control requires “air traffic control” orchestrated by the spin rate imposed by the “hub” of the spindle motor.

If the author is claiming to present the design principles of large-scale integration, the abstract and introduction should clearly state this and also some results should be shown. The title reminds me of the classic Science cover article of VLSI microfluidics of two layer pneumatic valves, Quake group, v 298, 5593, 2002. The emphasis on multiplexing and LSI throughout the article raises the expectation of the reader to see such an example.

The notion of fluidic larger-scale integration (LSI) has indeed been introduced about 20 years by the Quake group. We included citations of this pioneering work, and stressed that the presented work substantially distinguishes by the underlying microfluidic technology, the digital twin approach considering tolerances in experimental input parameters, and the modelling of various multiplexed (centrifugal) flow control schemes. I do not think that the Quake group intended to put a general and exclusive claim on the term LSI in view of the huge variety of microfluidic devices and platforms.

Instead, the article discusses the operation principles of LoDs and most commonly used valving mechanisms. In its current form, this article is much more suitable as a review article. Some parts of the article (especially introduction and conclusion) reads more like an editorial than a research article. Some buzzwords from other disciplines are used that makes the whole article much more fun to read. However, it does not fit to a research article since it does not give any relevant information. In some cases (such as the title, the digital twin analogy and the last paragraph on AI, Big Data, IoT, crowdsourcing, blockchain, etc..) these analogies are distracting rather than being helpful. I think the last paragraph is completely out of context despite the fact that it is fun to read. I also disagree with the digital twin analogy that was used throughput the article. The principles and guidelines given here are better represented as well-known engineering design principles. The digital twin analogy has a very different connotation than the ideas presented in this article.

The part the respected reviewer refers to is the outlook at the very end, which is well separated from the technical content. In this outlook, I drafted my own plans to launch a participatory research model within the described, admittedly unconventional framework. The references to the published articles supplied in this outlook provide further detail on the concept for implementation.

Overall, what is informative is the discussion on multiplexing section. Although, it presents the well-known facts of yield calculation and filter design principles used in micromanufacturing and circuit design, the discussion is helpful for integration of multiple valves and a LoD assay flow design. The spatial and frequency domain representations are very helpful in conveying the author's point (Fig 6). A conceptual assay example is given. In the last section, the hand-shake and triggering analogies explains the principle. Fig. 9 nicely illustrates the benefit of vented channel design.

Thanks for this. Indeed, the principles for the impact of accuracy and precision are known in general from other engineering disciplines. Yet, to my best knowledge, have not been applied to centrifugal systems, and the work presented underpins that these effects can, by no means, be neglected when aiming for LSI and system-level reliability at the same time. Furthermore, I have not been able to find articles referring to the intimate link between these tolerances, for instance, owing to machining and pipetting techniques, and the resulting “crowding” of the frequency space due to their finite band width enabled by the “digital twin” approach.

To me such a manuscript is much more suitable as a review article with a more focused title, abstract, introduction and conclusion. If the author prefers a research article, the demonstration of these principles as a LSI LoD demonstration is almost a necessity. Then, also the new information in this article should be clearly differentiated than the existing literature, preferably using a more technical and dull language than the author prefers.

Thank you for this valuable comment. While I respectfully disagree with the reviewer, his/her point revealed that the novelty of the research presented in the manuscript needed to be pointed out in a more compelling way. Accordingly, the abstract and final summary were substantially updated to point out that the manuscript develops a digital twin for several, previously published technologies for centrifugal flow control which, for the first time,

  • Mathematically connects retention rates of rotationally actuated valves to experimental parameters
  • Illustrates the concept of band widths in frequency space as the main quantity to watch for achieving operationally robust flow control
  • Introduces a set of quantitative performance metrics which allows to improve valving performance and disc designs for optimum use of real and frequency space as a pivotal prerequisite for automating multiplexed bioassay protocols
  • Elucidates the mathematical principles of several key mechanisms underlying advanced flow control

So, from my point of view, I would contend that these achievements, while evidently on the mathematical level, are novel and step significantly beyond the state of the art, and thus represent key attributes of an original research article.
From my point of view, inclusion of a practical demonstration is not suitable. On the one hand, it would only show one particular case amongst the myriads of possible scenarios in the multi-dimensional parameter space a reader might be interested in. It is way more useful to give the reader the digital twin at hand to address their specific use case. On the other hand, and in particular in academic environments which commonly cannot entertain professional, high-quality mass manufacturing facilities, it is simply not feasible to accumulate the statistics over sufficiently large ensembles of fluidic LoaD devices.

Thank you for this very original article and I hope you take my comments as constructively and find them useful in revising the work. I recommend a major revision of the article mostly due to the structural comments I had above. As a reader I would personally enjoy such a text very much as a review article.

Overall, despite some different points of view on the certain, somewhat arguable aspects, I very much thank the reviewer for his/her very well founded, highly qualified and constructive feedback, and his / her benevolent, interspersed encouragements!